# Study on Hematological and Biochemical Characters of Cloned Duroc Pigs and Their Progeny

**DOI:** 10.3390/ani9110912

**Published:** 2019-11-02

**Authors:** Ting Gu, Junsong Shi, Lvhua Luo, Zicong Li, Jie Yang, Gengyuan Cai, Enqin Zheng, Linjun Hong, Zhenfang Wu

**Affiliations:** 1National Engineering Research Center for Breeding Swine Industry, Guangdong Provincial Key Lab of Agro-Animal Genomics and Molecular Breeding, South China Agricultural University, Guangzhou 510642, China; tinggu@scau.edu.cn (T.G.); lizicong@scau.edu.cn (Z.L.); jieyang@scau.edu.cn (J.Y.); cgy0415@scau.edu.cn (G.C.); eqzheng@scau.edu.cn (E.Z.); linjun.hong@scau.edu.cn (L.H.); 2Wens Foodstuff Group Co., Ltd., Yunfu 527400, China; junsongstone@wens.com.cn (J.S.); lvye345@163.com (L.L.)

**Keywords:** pig, clones, progeny, blood, preserve, superior boars, food safety

## Abstract

**Simple Summary:**

Cloning is the most promising technique for passing the excellent phenotypes of the best individuals in the population. Here we studied the effects of cloning on Duroc pig, which is the most popular sire used in pig production due to its good growth and meat quality. Understanding the changes of cloned Duroc pigs and their progenies is of great importance for animal breeding and public acceptance. The results of this study suggested that there were no difference in blood parameters between the cloned Duroc and the conventionally bred Duroc and their progenies.

**Abstract:**

To increase public understanding in cloned animals produced by somatic cell nuclear transfer technology, our previous study investigated the carcass trait and meat quality of the clones (paper accepted), and this study we further evaluate differences by investigating the blood parameters in cloned pigs and their progeny. We collected blood samples from the clones and conventionally bred non-clones and their progeny, and investigated their hematological and blood biochemical characters. Our results supported the hypothesis that there was no significant difference between clones and non-clones, or their progeny. Taken together, the data demonstrated that the clones or their progeny were similar with their controls in terms of blood parameters, although there were still other kinds of disorders, such as abnormal DNA methylation or histone modifications that needs further investigation. The data in this study agreed that cloning technique could be used to preserve and enlarge the genetics of the superior boars in pig breeding industry, especially in facing of the deadly threat of African Swine fever happened in China.

## 1. Introduction

The first mammal cloned from an adult’s somatic cell succeeded in 1996, when Dolly the sheep was born in the Roslin Institute in Scotland [1]. Since then, somatic cell nuclear transfer technique (SCNT, also called cloning) is more and more widely applied in multiplying elite animals, conservation of endangered species, and assisting producing the genetically modified animals [2,3,4]. However, most countries have not approved the products, including meat and milk entering the food supply due to the uncertainty of the safety and health status of the cloned animals produced by SCNT technique [5].

Several studies indicated that the SCNT animals were different to the conventionally bred ones, such as significantly higher occurrence of malformation [6], higher whole genome-wide DNA methylation in the SCNT embryos [7], and aberrant methylated DNA in imprinted genes such as insulin like growth factor 2 (IGF2), H19, and X inactive specific transcript (XIST) [8], and shorter lifespan occurred [9]. Although other studies showed that the deficiency in embryonic reprogramming could be erased by the frequent embryonic loss or corrected during germline transmission [7,10], the healthy status of the cloned animals and the safety of the cloned animal’s products were still the concerns of the public as most countries have not explicitly allowed or refused the animal products from cloned animals and their progeny to enter the food supply markets [5]. Previously, we compared the carcass traits and biochemical constitution of the meat produced by cloned and conventionally bred pigs (paper accepted). Here we focused on the biochemistrical and hematological characters of their blood. Hematological cells, such as white blood cells (WBCs, also named leukocytes), red blood cells (RBCs, also named erythrocytes), and platelets are involved in the immune response process [11,12]. Leukocytes are involved in innate and adaptive immunity [13,14]. Erythrocytes are the transporters for oxygen, carbon dioxide and also involved in killing pathogens [15,16]. The aberrant RBCs number indicated the increasing risk of anemia, polycythemia, hypertension, and heart failure [17]. Platelets are small anucleate blood elements that involved in hemostasis, thrombosis, and immune response [18,19]. Therefore, the hematological parameters are important indicators of the general health status for human and animals as well, although some of the abnormal epigenetic modification mentioned above, such as DNA methylation and histone modification changes are not obligatory reflected by the blood parameters. Similar to the hematological parameters, the biochemical parameters are also indicators for the metabolism and healthy status of the animals [20,21].

In this study, we investigated the hematological and blood biochemical characters of the cloned and conventionally bred Duroc pigs, which is the most popular sire line in the pig breeding industry and we also collected samples of their crossbred and purebred progeny as well to figure out whether the blood parameters was changed by SCNT manipulation. The data obtained in this study will be helpful for public reliability of edible products derived from cloned animals and their progeny and also giving inspiration in further human medical therapy by transplantation of cloned organs.

## 2. Materials and Methods

### 2.1. Ethics Statement

All animals used in this study were reared and euthanized according to the Regulations for the Administration of Affairs Concerning Experimental Animals (Ministry of Science and Technology, China, revised June 2004) [22] and approved by the Animal Care Committee of South China Agricultural University, Guangzhou, China (approval number # SYXK2014-0136) [23].

### 2.2. Animals

There are three groups of comparisons in our study: Clones versus non-clones, F1 purebred progeny of clones versus non-clones (F1 purebred group), and F1 crossbred progeny of clones versus non-clones (F1 crossbred group). We obtained somatic cells for nuclear transferring from the fibroblast cell line derived from the ear of a superior purebred Duroc boar. Approximately 2000 reconstructed embryos were obtained by injecting the donor nuclei of the somatic cells into the enucleated matured oocytes cultured in vitro, and then surgically transferred into the oviducts of 10 anesthetized recipient gilts at two days after onset of estrus described previously [24]. Four clones were generated and housed in the same barn under identical feed and housing conditions with four naturally bred Duroc boars since weaning. Their blood was collected via ear vein at 72 weeks of age. Four F1 purebred progeny of the clones were generated by AI into naturally bred Duroc swine with semen derived from the clones in the group above and then housed together with four non-cloned progeny since weaning. Blood was collected at about 19 weeks of age from F1 purebred progeny of clones and their control. Six F1 crossbred cloned progeny were produced by AI into naturally bred Landrace swine with semen derived from the clones and then housed together with six non-cloned crossbred progeny from weaning. Blood was collected at about 180 days.

### 2.3. Blood Sample Collection and Analyses

After 12 h of fasting, whole blood was taken by the ear vein into tubes with or without anticoagulant heparin sodium (20 U/mL). Blood samples were profiled in Guangdong Medical Laboratory Animal center in Guangdong, China [25,26]. The hematological variables evaluated in our study were total white blood cell count and differential WBC count, including neutrophilic granulocyte (NE), lymphocyte (LY), monocyte (MO), eosinophil (EO), basophil (BA), red blood cells count, hemoglobin count (HGB), hematocrit value (HCT), mean corpuscular volume (MCV), mean corpuscular hemoglobin (MCH), mean corpuscular hemoglobin concentration (MCHC), RBC distribution width (RDW), platelet count (PLT), plateletcrit (PCT), mean platelet volume (MPV), and PLT volume distribution width (PDW). Twenty-four biochemical variables evaluated in our study were alanine transaminase (ALT), aspartate transaminase (AST), alkaline phosphatase (ALP), Creatine Kinase (CK), total protein (TP), albumin (ALB), globulins (GLOB), albumin to globulin ratio (A/G), glucose (GLU), blood urinary nitrogen (BUN), total cholesterol (TC), triglyceride (TG), high-density lipoprotein cholesterol (HDL-C), low-density lipoprotein cholesterol (LDL-C), uric acid (UA), lactate dehydrogenase (LDH), inorganic phosphorus (Pi), calcium (Ca), potassium (K+), sodium (Na+), chlorion (Cl−), iron (Fe), zinc (Zn), and testosterone (T).

### 2.4. Statistical Analysis

All the data was processed by Shapiro-Wilk goodness-of-fit test (SW or W test) for normality distribution fitness. Student’s *t*-test or Wilcoxon test were employed for the parameters that following or not following the normal distribution in each group (clones versus non-clones; F1 purebred group; and F1 crossbred group). All the data were presented in the form of mean ± standard error.

## 3. Results

### 3.1. Hematology

In this study, blood samples from each group were collected and the hematological parameters, such as WBC, RBC, and platelet and their related ratios were tested. All the parameters tested were within the reference range and about 90% of the data were following the normal distribution. There were no significant differences in most of the hematological parameters between the clones and their counterparts, although several exceptions existed, such as the differences in NE, NE%, and MO% were significant at 0.05 level in F1 crossbred group, while the differences in BA content, BA% were significant at 0.05 level and in MCH were significant at 0.01 level in between clones and their counterparts. There was no significant difference in hematological parameters in F1 purebred group (Table 1).

### 3.2. Blood Biochemistry

The biochemistrical parameters, including protein, glucose and lipid metabolism indicators, and content of metal ions were tested. There were no significant differences in the majority of the parameters detected, with the exception that the difference in GLOB and Pi were significant in between cloned and non-cloned pigs, while the differences in GLU, HDL-C were significant in between F1 purebred group. There was no significant difference in hematological parameters in between F1 crossbred group (Table 2).

To detect whether the cloning manipulation improved the uniformity between individuals, we calculated the relative coefficient value of blood biochemistry parameters of clones with their counterparts. Interestingly, reduced variation in clones and their progeny was only observed in several parameters, while variations in CK, TP, GLOB, GLU, BUN, UA, Pi, and Na+ seems greater than their counterparts. Notably, the variation in crossbred progeny seems lower than the clones and the purebred progeny (Figure 1). Similar lower variation results were observed in the crossbred progeny in hematological parameters as well (Figure 2).

## 4. Discussion

Duroc is one of the most popular sire breeds in pig breeding. It is excellent in growth rate, body composition, and it is famous for high intramuscular fat content in meat [27,28]. Selection for the best sire individuals is a most time and money consuming work in pig breeding, and furthermore, how to preserve and make full use of this super boar after picking up the best sire in the population is another question. SCNT is a potential solution for the question, which still needs further investigation in details.

Previous studies indicated that epigenetic modifications were altered in cloned abnormal animals compared to naturally bred ones. Global hypo-acetylation, reduced DNA methylation in satellite I region, loss of imprinting (LOS), reduced histone methylation were reported in the somatic cell nuclear transfer derived animals [29,30,31]. Piglets derived by SCNT also showed a higher incidence of malformations and a higher death rate during the perinatal period [32]. To assess the safety and health status of cloned animal and their progeny, comparisons were carried out in the hematological and biochemical parameters. Our results showed that there were no significant differences between clones and their counterpart for more than 40 parameters, as well as the purebred and crossbred progeny. Our results in clones and their crossbred and purebred progeny were in consistent with the previous reports in cloned cattle and pig, as differences in several biochemical parameters were observed in heifers and the F1 progeny of clones although the bred of the pigs was not mentioned [33,34]. Furthermore, the variability of the crossbred progeny of clones seems less than the clones and their purebred progeny, although further studies are still needed. This generation “correction” was in line with the previous report although the breed of cloned pigs was not mentioned, in that study several parameters including Creatine, ALP, phosphorus, calcium, and BUN were significantly less variable in clones than the control pigs at both 15 and 27 weeks [35], and these significantly difference were “corrected” in the offspring of clones with the exception of BUN and ALP [33]. In our study, the variation in clones and the control pigs did not show significant difference in BUN nor ALP (Table 2), which may cause by the difference of the pig breed.

## 5. Conclusions

The results showed for the first time that blood parameters were similar in cloned Duroc boar and the F1 purebred and crossbred progeny to their respective controls, which provided more information in the concerning of using the cloning technique in enlarging the superior sire for pig breeding.

## Figures and Tables

**Figure 1 animals-09-00912-f001:**
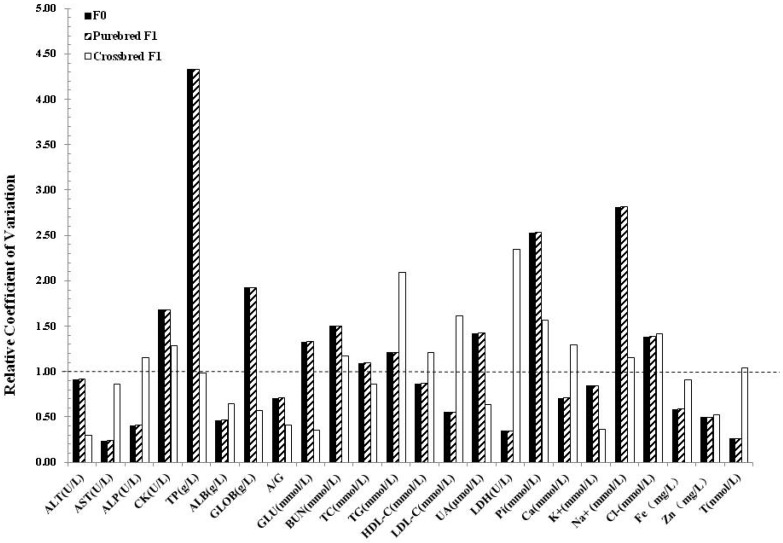
Variability of blood biochemistry parameters in clones and their progenies. The relative coefficient of variation (C.V.) was calculated by the C.V. of cloned pigs divided by their counterparts.

**Figure 2 animals-09-00912-f002:**
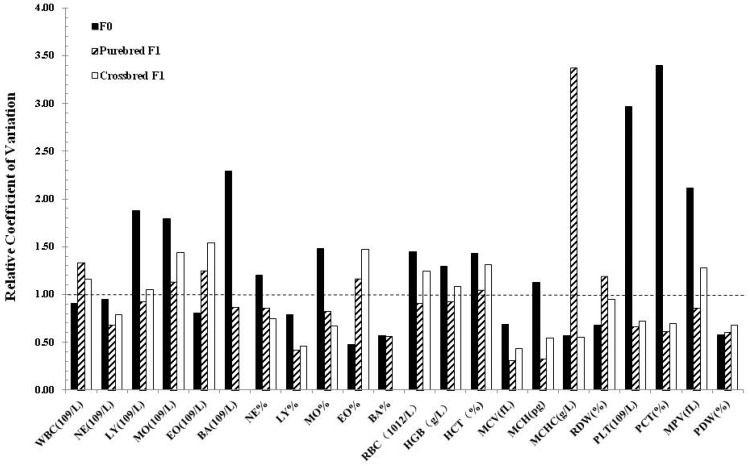
Variability of hematological variables in clones and their progenies. The relative coefficient of variation was calculated by the C.V. of cloned pigs divided by their counterparts.

**Table 1 animals-09-00912-t001:** Hematological variables for blood from cloned and non-cloned pigs and their purebred and crossbred progenies.

Hematological Variables	Clones (*n* = 4)	Non-clones (*n* = 4)	F1 Clone x Non-clone (*n* = 4)	F1 Non-clone x Non-clone (*n* = 4)	Clone x Landrace Crossbreeds Cloned (*n* = 6)	Non-clone x Landrace Crossbreeds (*n* = 6)
WBC (10^9^/L)	13.53 ± 2.64	13.24 ± 2.83	22.87 ± 4.40	22.14 ± 3.19	17.08 ± 1.79	20.67 ± 1.86
NE (10^9^/L)	5.35 ± 2.05	7.07 ± 2.84	6.45 ± 1.31	9.61 ± 2.86	4.63 ± 0.52 ^a^	7.93 ± 1.13 ^b^
LY (10^9^/L)	7.53 ± 1.44	5.74 ± 0.58	15.64 ± 3.81	11.91 ± 3.11	12.03 ± 1.32	12.35 ± 1.30
MO (10^9^/L)	0.27 ± 0.33	0.11 ± 0.07	0.33 ± 0.32	0.08 ± 0.06	0.22 ± 0.04	0.20 ± 0.03
EO (10^9^/L)	0.31 ± 0.08	0.28 ± 0.08	0.22 ± 0.12	0.46 ± 0.21	0.20 ± 0.05	0.18 ± 0.03
BA (10^9^/L)	0.08 ± 0.02 ^a^	0.04 ± 0.01 ^b^	0.23 ± 0.13	0.08 ± 0.05	0	0
NE%	38.85 ± 9.69	51.85 ± 10.7	28.75 ± 6.76	43.50 ± 11.84	27.50 ± 2.22 ^a^	38.12 ± 4.11 ^b^
LY%	56.13 ± 9.61	44.75 ± 9.69	68.00 ± 6.32	53.68 ± 11.83	70.12 ± 2.19	60.02 ± 4.07
MO%	2.18 ± 2.67	0.875 ± 0.72	1.38 ± 1.15	0.33 ± 0.33	1.30 ± 0.11 ^a^	0.90 ± 0.11 ^b^
EO%	2.28 ± 0.49	2.20 ± 1.00	0.95 ± 0.53	2.13 ± 1.01	1.08 ± 0.20	0.97 ± 0.12
BA%	0.58 ± 0.13 ^a^	0.33 ± 0.12 ^b^	0.92 ± 0.39	0.38 ± 0.28	0	0
RBC (10^12^/L)	7.88 ± 0.68	7.59 ± 0.45	7.61 ± 0.37	7.51 ± 0.41	6.12 ± 0.33	6.33 ± 0.27
HGB (g/L)	153.75 ± 13.23	141.00 ± 9.38	126.00 ± 5.16	121.00 ± 5.35	114.17 ± 5.31	112.50 ± 4.81
HCT (%)	49.90 ± 4.66	46.48 ± 3.03	43.18 ± 1.74	41.10 ± 1.59	32.62 ± 1.51	32.62 ± 1.14
MCV (fL)	63.33 ± 1.25	61.25 ± 1.76	56.85 ± 1.08	54.83 ± 3.29	53.35 ± 0.60	51.67 ± 1.37
MCH (pg)	19.53 ± 0.32 ^A^	18.60 ± 0.27 ^B^	16.63 ± 0.34	16.13 ± 0.99	18.68 ± 0.29	17.80 ± 0.51
MCHC (g/L)	308.25 ± 5.32	304.25 ± 9.17	292.50 ± 5.80	294.50 ± 1.73	350.00 ± 2.13	344.50 ± 3.79
RDW (%)	18.45 ± 0.51	18.53 ± 0.74	19.83 ± 1.49	20.80 ± 1.32	16.05 ± 0.25	15.43 ± 0.25
PLT (109/L)	151.25 ± 82.66	204.75 ± 37.73	252.50 ± 21.39	218.25 ± 27.67	439.17 ± 0.396	367.33 ± 61.34
PCT (%)	0.16 ± 0.09	0.21 ± 0.03	0.24 ± 0.03	0.20 ± 0.04	0.21 ± 0.03	0.19 ± 0.03
MPV (fL)	10.40 ± 0.74	10.30 ± 0.34	9.45 ± 0.70	9.28 ± 0.80	4.82 ± 0.15	4.98 ± 0.12
PDW (%)	15.65 ± 0.17	15.30 ± 0.29	15.23 ± 0.22	14.90 ± 0.36	16.93 ± 0.30	16.80 ± 0.45

Note: *n* = number of animals in each group. Results are presented as mean ± S.E. Means with different lower superscripts (a and b in the table) indicated difference was significant at 0.05 level within the group of comparison underlined with solid line or dotted lines (*p* < 0.05), while means with upper superscripts (A and B in the table) indicated difference was significant at 0.01 level within the group of comparison underlined with solid line or dotted lines (*p* < 0.01). Abbreviations in the tables were listed below: WBC: Total white blood cell; NE: Neutrophilic granulocyte; LY: Lymphocyte; MO: Monocyte; EO: Eosinophil; BA: Basophil; RBC: Red blood cells count; HGB: Hemoglobin count; HCT: Hematocrit value; MCV: Mean corpuscular volume; MCH: Mean corpuscular hemoglobin; MCHC: Mean corpuscular hemoglobin concentration; RDW: RBC distribution width; PLT: Platelet count; PCT: Plateletcrit; MPV: Mean platelet volume; and PDW: PLT volume distribution width.

**Table 2 animals-09-00912-t002:** Biochemistry parameters for blood from clones and non-clones and their purebred and crossbred progenies.

Biochemical Parameters	Clones (n = 4)	Non-clones (n = 4)	F1 Clone x Non-clone (n = 4)	F1 Non-clone x Non-clone (n = 4)	Clone x Landrace Crossbreeds (n = 6)	Non-clone x Landrace Crossbreeds (n = 6)
ALT (U/L)	43.75 ± 7.14	43.75 ± 7.80	39.50 ± 8.35	35.75 ± 6.95	41.33 ± 1.43	35.67 ± 4.17
AST (U/L)	39.25 ± 4.50	54.50 ± 26.38	27.50 ± 5.45	35.75 ± 12.89	44.83 ± 6.17	46.83 ± 7.49
ALP (U/L)	68.00 ± 4.76	52.25 ± 8.99	128.25 ± 21.36	150.75 ± 33.01	63.83 ± 5.55	81.33 ± 6.15
CK (U/L)	1777.25 ± 866.34	1206.75 ± 350.27	1190.00 ± 470.34	1379.00 ± 963.95	1641.17 ± 620.39	1395.33 ± 411.63
TP (g/L)	82.70 ± 4.20	76.38 ± 0.90	67.25 ± 4.86	66.00 ± 2.29	72.87 ± 1.92	71.87 ± 1.93
ALB (g/L)	40.98 ± 0.92	40.08 ± 1.96	34.18 ± 3.14	34.40 ± 2.24	29.62 ± 0.90	28.25 ± 1.34
GLOB (g/L)	41.73 ± 3.37 ^a^	36.30 ± 1.52^b^	33.08 ± 2.39	31.60 ± 2.10	43.25 ± 1.63	43.62 ± 2.86
A/G	0.99 ± 0.06	1.11 ± 0.09	1.03 ± 0.09	1.09 ± 0.12	0.63 ± 0.03	0.67 ± 0.07
GLU (mmol/L)	3.19 ± 0.66	3.35 ± 0.52	4.52 ± 0.40^a^	5.31 ± 0.27 ^b^	3.02 ± 0.34	3.62 ± 0.15
BUN (mmol/L)	5.48 ± 0.95	5.37 ± 0.62	4.00 ± 1.39	3.57 ± 1.09	4.61 ± 0.27	5.21 ± 0.35
TC (mmol/L)	1.30 ± 0.11	1.37 ± 0.11	2.02 ± 0.13	1.99 ± 0.18	3.57 ± 0.19	3.23 ± 0.20
TG (mmol/L)	0.28 ± 0.12	0.22 ± 0.07	0.38 ± 0.06	0.29 ± 0.08	0.45 ± 0.08	0.45 ± 0.04
HDL-C (mmol/L)	0.65 ± 0.08	0.58 ± 0.08	0.85 ± 0.09 ^a^	0.68 ± 0.08 ^b^	1.21 ± 0.08	1.23 ± 0.07
LDL-C (mmol/L)	0.51 ± 0.04	0.63 ± 0.09	1.00 ± 0.04	1.13 ± 0.15	1.56 ± 0.20	1.38 ± 0.11
UA (µmol/L)	1.75 ± 0.95	1.50 ± 0.57	1.25 ± 0.50	1.50 ± 1.00	5.67 ± 0.48	5.17 ± 0.48
LDH (U/L)	435.50 ± 65.23	493.50 ± 212.64	491.25 ± 48.33	464.25 ± 81.88	851.33 ± 97.83	675.00 ± 33.02
Pi (mmol/L)	2.34 ± 0.08 ^A^	2.15 ± 0.03 ^B^	3.21 ± 0.17	2.72 ± 0.34	3.31 ± 0.09	3.17 ± 0.05
Ca (mmol/L)	2.68 ± 0.02	2.65 ± 0.03	2.72 ± 0.09	2.69 ± 0.13	2.61 ± 0.05	2.50 ± 0.04
K+ (mmol/L)	4.56 ± 0.23	4.54 ± 0.27	5.29 ± 0.77	5.30 ± 1.01	4.78 ± 0.13	4.75 ± 0.34
Na+ (mmol/L)	142.25 ± 6.19	145.00 ± 2.24	145.48 ± 4.70	145.33 ± 3.78	140.90 ± 2.62	137.98 ± 2.22
Cl− (mmol/L)	97.00 ± 3.12	95.10 ± 2.21	99.08 ± 2.56	100.70 ± 4.20	96.95 ± 1.73	98.73 ± 1.25
Fe (mg/L)	5.45 ± 2.69	5.10 ± 4.28	3.80 ± 1.98	6.43 ± 4.01	17.56 ± 1.06	19.66 ± 1.31
Zn (mg/L)	0.76 ± 0.10	0.65 ± 0.18	0.41 ± 0.14	0.46 ± 0.30	9.14 ± 0.32	9.84 ± 0.65
T (nmol/L)	24.44 ± 2.12	27.54 ± 9.13	38.59 ± 5.57	26.65 ± 7.72	23.07 ± 2.88	20.19 ± 2.88

Note: Results are presented as mean ± S.E. Means with different lower superscripts (a and b in the table) indicated difference was significant at 0.05 level within the group of comparison underlined with solid line or dotted lines (*p* < 0.05), while means with upper superscripts (A and B in the table) indicated difference was significant at 0.01 level within the group of comparison underlined with solid line or dotted lines (*p* < 0.01). Abbreviations in the tables were listed below: ALT: Alanine transaminase; AST: Aspartate transaminase; ALP: Alkaline phosphatase; CK: Creatine Kinase; TP: Total protein; ALB: Albumin; GLOB: globulins; A/G: Albumin to globulin ratio; GLU: Glucose; BUN: Blood urinary nitrogen; TC: Total cholesterol; TG: Triglyceride; HDL-C: High-density lipoprotein cholesterol; LDL-C: Low-density lipoprotein cholesterol; UA: Uric acid; LDH: Lactate dehydrogenase; Pi: Inorganic phosphorus; Ca: Calcium; K+: Potassium; Na+: Sodium; Cl−: Chlorion; Fe: Iron; Zn: Zinc; and T: Testosterone.

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
