# Peer review of "Study on Hematological and Biochemical Characters of Cloned Duroc Pigs and Their Progeny"

_animals, 2019, doi:10.3390/ani9110912_

Round 1
Reviewer 1 Report
the authors addressed the concerns raised in the original review appropriately and the manuscript can now be accepted
Reviewer 2 Report
Thank you. I am satisfied with the corrections, though some minor grammar errors remain.
This manuscript is a resubmission of an earlier submission. The following is a list of the peer review reports and author responses from that submission.
Round 1
Reviewer 1 Report
well- written manuscript evaluating the hematological and biochemical characteristics
why was blood obtained at various time points from different sets of comparison swine - i.e. 72 weeks, 19 weeks and 180 days? The reviewer realizes that the SCNT boars were much older than the offspring but a consistent time point on the comparison of progeny may would have made for a stronger paper as the ones at 180 days would be under the influence of pubertal hormonal profile than those closer to 19 weeks. Unsure if this would have changed any parameters but something to consider.
no other modifications needed
Author Response
Response: The comparisons were made within the three groups between “cloned” and “non-cloned” related: clones vs non-clones, F1 purebred progeny of clones vs non-clones (F1 purebred group), and F1 crossbred progeny of clones vs non-clones (F1 crossbred group), and the further comparison in between the groups was made by an indicator called “relative coefficient variation” (C.V.), which divided the C.V. of clone-related by their age-matched compartments, so the age differences is not that important in our design, although we agreed that a consistent time point would be better. To eliminate the effects, such as seasons, temperatures and feed in the blood parameters, the blood were collected at the similar time and sent to analysis at one time, so the SCNT boars were much older than the progeny.

Reviewer 2 Report
The sample size appears small. Can you justify the n of 4 per sample?
It is not clear to me what “non-cloned” means. Are these the normal offspring of the same, one boar you cloned from? You need to elaborate on: 1) how many boars were cloned, how this was done, into how many sows, and when. 2) What pigs were compared? The clones were compared to the boars they were cloned from? What is the source of the “non-cloned” boars?
Table 1 is partially cut-off. It also is not clear. “Cloned” grammatically implies the source of the material from which the clone was produced, not the clones themselves. Please rename the columns to be more clear: Clones, non-clones, F1 clone x non-clone, F1 non-clone x non-clone, clone x Landrace crossbreeds.
The abbreviations in Table 1 need to be defined somewhere.
The first paragraph of the Discussion sounds more like an Introduction paragraph.
In line 154 you say “for the first time,” but earlier you
The language needs editing. A pass by a native English speaker should be enough.
For example:
4 - “technique for passing”
8 -“results”
10 - delete “also”
12 - delete “(also called clone)”
13- Delete “Here”
14 delete “as well” and “then”
16 replace “as well as their” with “or”
18- Split into two sentences.
19 “superior” ?
22 “from an adult’s”
24 “cloning”
25 “production of genetically modified”
27 delete “although they did not forbid them”
29 - delete “by a two-year worldwide survey”
Author Response
It is not clear to me what “non-cloned” means. Are these the normal offspring of the same, one boar you cloned from? You need to elaborate on: 1) how many boars were cloned, how this was done, into how many sows, and when. 2) What pigs were compared? The clones were compared to the boars they were cloned from? What is the source of the “non-cloned” boars?
Response: The “non-cloned” means the normal offspring of the same boar we cloned from.
We obtained somatic cells for nuclear transferring from the fibroblast cell line derived from the ear of a superior purebred Duroc boar. Approximately 2000 reconstructed embryos were obtained by injecting the donor nuclei of the somatic cells into the enucleated matured oocytes cultured in vitro, and then surgically transferred into the oviducts of 10 anesthetized recipient gilts at 2 days after onset of estrus described previously [24]. The clones were compared to the “non-clones”, boars that produced by artificial insemination with the semen derived from the donor boars, and then the clones and non-clones were born at similar time and then housed under the same condition.We have added these details in the “materials and methods”.
Table 1 is partially cut-off. It also is not clear. “Cloned” grammatically implies the source of the material from which the clone was produced, not the clones themselves. Please rename the columns to be more clear: Clones, non-clones, F1 clone x non-clone, F1 non-clone x non-clone, clone x Landrace crossbreeds.
Response: We have adjusted the Table 1. We have renamed the columns according to reviewer’s suggestion.
The abbreviations in Table 1 need to be defined somewhere.
Response: We have listed the abbreviations under the tables.
The first paragraph of the Discussion sounds more like an Introduction paragraph.
In line 154 you say “for the first time,” but earlier you
Response: It is the first time to systematically study the hematological and biochemistrical parameters of blood from the clones and their purebred and crossbred progeny. We have deleted the “the first time” anyway.
The language needs editing. A pass by a native English speaker should be enough.
For example:
4 - “technique for passing”
8 -“results”
10 - delete “also”
12 - delete “(also called clone)”
13- Delete “Here”
14 delete “as well” and “then”
16 replace “as well as their” with “or”
18- Split into two sentences.
19 “superior” ?
22 “from an adult’s”
24 “cloning”
25 “production of genetically modified”
27 delete “although they did not forbid them”
29 - delete “by a two-year worldwide survey”
Response: We have adjusted all the places involved according to the reviewer’s suggestions.
